# Deep Network Classification by Scattering and Homotopy Dictionary Learning

**John Zarka, Louis Thiry, Tomás Angles**
Département d'informatique de l'ENS, ENS, CNRS, PSL University, Paris, France
`{john.zarka,louis.thiry,tomas.angles}@ens.fr`

**Stéphane Mallat**
Collège de France, Paris, France
Flatiron Institute, New York, USA

## Abstract

We introduce a sparse scattering deep convolutional neural network, which provides a simple model to analyze properties of deep representation learning for classification. Learning a single dictionary matrix with a classifier yields a higher classification accuracy than AlexNet over the ImageNet 2012 dataset. The network first applies a scattering transform that linearizes variabilities due to geometric transformations such as translations and small deformations. A sparse $\ell^1$ dictionary coding reduces intra-class variability while preserving class separation through projections over unions of linear spaces. It is implemented in a deep convolutional network with a homotopy algorithm having an exponential convergence. A convergence proof is given in a general framework that includes ALISTA. Classification results are analyzed on ImageNet.

## 1 Introduction

Deep convolutional networks have spectacular applications to classification and regression (LeCun et al., 2015), but they are black boxes that are hard to analyze mathematically because of their architecture complexity. Scattering transforms are simplified convolutional neural networks with wavelet filters which are not learned (Bruna & Mallat, 2013). They provide state-of-the-art classification results among predefined or unsupervised representations, and are nearly as efficient as learned deep networks on relatively simple image datasets, such as digits in MNIST, textures (Bruna & Mallat, 2013) or small CIFAR images (Oyallon & Mallat, 2014; Mallat, 2016). However, over complex datasets such as ImageNet, the classification accuracy of a learned deep convolutional network is much higher than a scattering transform or any other predefined representation (Oyallon et al., 2019). A fundamental issue is to understand the source of this improvement. This paper addresses this question by showing that one can reduce the learning to a single dictionary matrix, which is used to compute a positive sparse $\ell^1$ code.

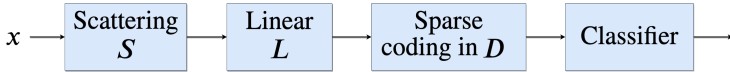

Figure 1: A sparse scattering network is composed of a scattering transform $S$ followed by an optional linear operator $L$ that reduces its dimensionality. A sparse code approximation of scattering coefficients is computed in a dictionary $D$. The dictionary $D$ and the classifier are jointly learned by minimizing the classification loss with stochastic gradient descent.

The resulting algorithm is implemented with a simplified convolutional neural network architecture illustrated in Figure 1. The classifier input is a positive $\ell^1$ sparse code of scattering coefficients calculated in a dictionary $D$. The matrix $D$ is learned together with the classifier by minimizing a classification loss over a training set. We show that learning $D$ improves the performance of a scattering

representation considerably and is sufficient to reach a higher accuracy than AlexNet (Krizhevsky et al., 2012) over ImageNet 2012. This cascade of well understood mathematical operators provides a simplified mathematical model to analyze optimization and classification performances of deep neural networks.

Dictionary learning for classification was introduced in Mairal et al. (2009) and implemented with deep convolutional neural network architectures by several authors (Sulam et al., 2018; Mahdizade-haghdam et al., 2019; Sun et al., 2018). To reach good classification accuracies, these networks cascade several dictionary learning blocks. As a result, there is no indication that these operators compute optimal sparse $\ell^1$ codes. These architectures are thus difficult to analyze mathematically and involve heavy calculations. They have only been applied to small image classification problems such as MNIST or CIFAR, as opposed to ImageNet. Our architecture reaches a high classification performance on ImageNet with only one dictionary $D$, because it is applied to scattering coefficients as opposed to raw images. Intra-class variabilities due to geometric image transformations such as translations or small deformations are linearized by a scattering transform (Bruna & Mallat, 2013), which avoids unnecessary learning.

Learning a dictionary in a deep neural network requires to implement a sparse $\ell^1$ code. We show that homotopy iterative thresholding algorithms lead to more efficient sparse coding implementations with fewer layers. We prove their exponential convergence in a general framework that includes the ALISTA (Liu et al., 2019) algorithm. The main contributions of the paper are summarized below:

- A sparse scattering network architecture, illustrated in Figure 1, where the classification is performed over a sparse code computed with a single learned dictionary of scattering coefficients. It outperforms AlexNet over ImageNet 2012.

- A new dictionary learning algorithm with homotopy sparse coding, optimized by gradient descent in a deep convolutional network. If the dictionary is sufficiently incoherent, the homotopy sparse coding error is proved to convergence exponentially.

We explain the implementation and mathematical properties of each element of the sparse scattering network. Section 2 briefly reviews multiscale scattering transforms. Section 3 introduces homotopy dictionary learning for classification, with a proof of exponential convergence under appropriate assumptions. Section 4 analyzes image classification results of sparse scattering networks on ImageNet 2012.

## 2 SCATTERING TRANSFORM

A scattering transform is a cascade of wavelet transforms and ReLU or modulus non-linearities. It can be interpreted as a deep convolutional network with predefined wavelet filters (Mallat, 2016). For images, wavelet filters are calculated from a mother complex wavelet $\psi$ whose average is zero. It is rotated by $r_{-\theta}$, dilated by $2^j$ and its phase is shifted by $\alpha$:

$$\psi_{j,\theta}(u) = 2^{-2j}\psi(2^{-j}r_{-\theta}u) \text{ and } \psi_{j,\theta,\alpha} = \text{Real}(e^{-i\alpha}\,\psi_{j,\theta})$$

We choose a Morlet wavelet as in Bruna & Mallat (2013) to produce a sparse set of non-negligible wavelet coefficients. A ReLU is written $\rho(a) = \max(a, 0)$.

Scattering coefficients of order $m = 1$ are computed by averaging rectified wavelet coefficients with a subsampling stride of $2^J$:

$$Sx(u, k, \alpha) = \rho(x \star \psi_{j,\theta,\alpha}) \star \phi_J(2^J u) \text{ with } k = (j, \theta)$$

where $\phi_J$ is a Gaussian dilated by $2^J$ (Bruna & Mallat, 2013). The averaging by $\phi_J$ eliminates the variations of $\rho(x \star \psi_{j,\theta,\alpha})$ at scales smaller than $2^J$. This information is recovered by computing their variations at all scales $2^{j'} < 2^J$, with a second wavelet transform. Scattering coefficients of order two are:

$$Sx(u, k, k', \alpha, \alpha') = \rho(\rho(x \star \psi_{j,\theta,\alpha}) \star \psi_{j',\theta',\alpha'}) \star \phi_J(2^J u) \text{ with } k, k' = (j, \theta), (j', \theta')$$

To reduce the dimension of scattering vectors, we define phase invariant second order scattering coefficients with a complex modulus instead of a phase sensitive ReLU:

$$Sx(u, k, k') = ||x \star \psi_{j,\theta}| \star \psi_{j',\theta'}| \star \phi_J(2^J u) \text{ for } j' > j$$

The scattering representation includes order 1 coefficients and order 2 phase invariant coefficients. In this paper, we choose $J = 4$ and hence 4 scales $1 \leq j \leq J$, 8 angles $\theta$ and 4 phases $\alpha$ on $[0, 2\pi]$. Scattering coefficients are computed with the software package Kymatio (Andreux et al., 2018). They preserve the image information, and $x$ can be recovered from $Sx$ (Oyallon et al., 2019). For computational efficiency, the dimension of scattering vectors can be reduced by a factor 6 with a linear operator $L$ that preserves the ability to recover a close approximation of $x$ from $LSx$. The dimension reduction operator $L$ of Figure 1 may be an orthogonal projection over the principal directions of a PCA calculated on the training set, or it can be optimized by gradient descent together with the other network parameters.

The scattering transform is Lipschitz continuous to translations and deformations (Mallat, 2012). Intra-class variabilities due to translations smaller than $2^J$ and small deformations are linearized. Good classification accuracies are obtained with a linear classifier over scattering coefficients in image datasets where translations and deformations dominate intra-class variabilities. This is the case for digits in MNIST or texture images (Bruna & Mallat, 2013). However, it does not take into account variabilities of pattern structures and clutter which dominate complex image datasets. To remove this clutter while preserving class separation requires some form of supervised learning. The sparse scattering network of Figure 1 computes a sparse code of scattering representation $\beta = LSx$ in a learned dictionary $D$ of scattering features, which minimizes the classification loss. For this purpose, the next section introduces a homotopy dictionary learning algorithm, implemented in a small convolutional network.

## 3 HOMOTOPY DICTIONARY LEARNING FOR CLASSIFICATION

Task-driven dictionary learning for classification with sparse coding was proposed in Mairal et al. (2011). We introduce a small convolutional network architecture to implement a sparse $\ell^1$ code and learn the dictionary with a homotopy continuation on thresholds. The next section reviews dictionary learning for classification. Homotopy sparse coding algorithms are studied in Section 3.2.

### 3.1 SPARSE CODING AND DICTIONARY LEARNING

Unless specified, all norms are Euclidean norms. A sparse code approximates a vector $\beta$ with a linear combination of a minimum number of columns $D_m$ of a dictionary matrix $D$, which are normalized $\|D_m\| = 1$. It is a vector $\alpha^0$ of minimum support with a bounded approximation error $\|D\alpha^0 - \beta\| \leq \sigma$. Such sparse codes have been used to optimize signal compression (Mallat & Zhang, 1993) and to remove noise, to solve inverse problems in compressed sensing (Candes et al., 2006), and for classification (Mairal et al., 2011). In this case, the dictionary learning optimizes the matrix $D$ in order to minimize the classification loss. The resulting columns $D_m$ can be interpreted as classification features selected by the sparse code $\alpha^0$. To enforce this interpretation, we impose that sparse code coefficients are positive, $\alpha^0 \geq 0$.

**Positive sparse coding** Minimizing the support of a code $\alpha$ amounts to minimizing its $\ell^0$ "norm", which is not convex. This non-convex optimization is convexified by replacing the $\ell^0$ norm by an $\ell^1$ norm. Since $\alpha \geq 0$, we have $\|\alpha\|_1 = \sum_m \alpha(m)$. The minimization of $\|\alpha\|_1$ with $\|D\alpha - \beta\| \leq \sigma$ is solved by minimizing a convex Lagrangian with a multiplier $\lambda_*$ which depends on $\sigma$:

$$\alpha^1 = \operatorname*{argmin}_{\alpha \geq 0} \frac{1}{2}\|D\alpha - \beta\|^2 + \lambda_* \|\alpha\|_1 \tag{1}$$

One can prove (Donoho & Elad, 2006) that $\alpha^1(m)$ has the same support as the minimum support sparse code $\alpha^0(m)$ along $m$ if the support size $s$ and the dictionary coherence satisfy:

$$s\,\mu(D) < 1/2 \text{ where } \mu(D) = \max_{m \neq m'} |D_m^t D_{m'}| \tag{2}$$

The sparse approximation $D\alpha^1$ is a non-linear filtering which preserves the components of $\beta$ which are "coherent" in the dictionary $D$, represented by few large amplitude coefficients. It eliminates the "noise" corresponding to incoherent components of $\beta$ whose correlations with all dictionary vectors $D_m$ are typically below $\lambda_*$, which can be interpreted as a threshold.

**Supervised dictionary learning with a deep neural network**  Dictionary learning for classification amounts to optimizing the matrix $D$ and the threshold $\lambda_*$ to minimize the classification loss on a training set $\{(x_i, y_i)\}_i$. This is a much more difficult non-convex optimization problem than the convex sparse coding problem (1). The sparse code $\alpha^1$ of each scattering representation $\beta = LSx$ depends upon $D$ and $\lambda_*$. It is used as an input to a classifier parametrized by $\Theta$. The classification loss $\sum_i \text{Loss}(D, \lambda_*, \Theta, x_i, y_i)$ thus depends upon the dictionary $D$ and $\lambda_*$ (through $\alpha^1$), and on the classification parameters $\Theta$. The dictionary $D$ is learned by minimizing the classification loss. This task-driven dictionary learning strategy was introduced in Mairal et al. (2011).

An implementation of the task-driven dictionary learning strategy with deep neural networks has been proposed in (Papyan et al., 2017; Sulam et al., 2018; Mahdizadehaghdam et al., 2019; Sun et al., 2018). The deep network is designed to approximate the sparse code by unrolling a fixed number $N$ of iterations of an iterative soft thresholding algorithm. The network takes $\beta$ as input and is parametrized by the dictionary $D$ and the Lagrange multiplier $\lambda_*$, as shown in Figure 2. The classification loss is then minimized with stochastic gradient descent on the classifier parameters and on $D$ and $\lambda_*$. The number of layers in the network is equal to the number $N$ of iterations used to approximate the sparse code. During training, the forward pass approximates the sparse code with respect to the current dictionary, and the backward pass updates the dictionary through a stochastic gradient descent step.

For computational efficiency the main issue is to approximate $\alpha^1$ with as few layers as possible and hence find an iterative algorithm which converges quickly. Next section shows that this can be done with homotopy algorithms, that can have an exponential convergence.

## 3.2 Homotopy Iterated Soft Thresholding Algorithms

Sparse $\ell^1$ codes are efficiently computed with iterative proximal gradient algorithms (Combettes & Pesquet, 2011). For a positive sparse code, these algorithms iteratively apply a linear operator and a rectifier which acts as a positive thresholding. They can thus be implemented in a deep neural network. We show that homotopy algorithms can converge exponentially and thus lead to precise calculations with fewer layers.

**Iterated Positive Soft Thresholding with ReLU**  Proximal gradient algorithms compute sparse $\ell^1$ codes with a gradient step on the regression term $\|x - Dz\|^2$ followed by proximal projection which enforces the sparse penalization (Combettes & Pesquet, 2011). For a positive sparse code, the proximal projection is defined by:

$$\text{prox}_\lambda(\beta) = \underset{\alpha \geq 0}{\text{argmin}} \, \frac{1}{2}\|\alpha - \beta\|^2 + \lambda \, \|\alpha\|_1 \tag{3}$$

Since $\|\alpha\|_1 = \sum_m \alpha(m)$ for $\alpha(m) \geq 0$, we verify that $\text{prox}_\lambda(\beta) = \rho(\beta - \lambda)$ where $\rho(a) = \max(a, 0)$ is a rectifier, with a bias $\lambda$. The rectifier acts as a positive soft thresholding, where $\lambda$ is the threshold. Without the positivity condition $\alpha \geq 0$, the proximal operator in (3) is a soft thresholding which preserves the sign.

An Iterated Soft Thresholding Algorithm (ISTA) (Daubechies et al., 2004) computes an $\ell^1$ sparse code $\alpha^1$ by alternating a gradient step on $\|Dx - z\|^2$ and a proximal projection. For positive codes, it is initialized with $\alpha_0 = 0$, and:

$$\alpha_{n+1} = \rho(\alpha_n + \epsilon D^t(\beta - D\alpha_n) - \epsilon\lambda_*) \ \text{ with } \ \epsilon < \frac{1}{\|D^t D\|_{2,2}} \tag{4}$$

where $\|\,.\,\|_{2,2}$ is the spectral norm. The first iteration computes a non-sparse code $\alpha_1 = \rho(\epsilon D^t \beta - \epsilon\lambda_*)$ which is progressively sparsified by iterated thresholdings. The convergence is slow: $\|\alpha_n - \alpha^1\| = O(n^{-1})$. Fast Iterated Soft Thresholding Agorithm (FISTA) (Beck & Teboulle, 2009) accelerates the error decay to $O(n^{-2})$, but it remains slow.

Figure 2: A generalized ISTC network computes a positive $\ell^1$ sparse code in a dictionary $D$ by using an auxiliary matrix $W$. Each layer applies $Id - W^t D$ together with a ReLU and a bias $\lambda_n$ to compute $\alpha_n$ from $\alpha_{n-1}$ in (6). The original ISTC algorithm corresponds to $W = D$.

Each iteration of ISTA and FISTA is computed with linear operators and a thresholding and can be implemented with one layer (Papyan et al., 2017). The slow convergence of these algorithms requires to use a large number $N$ of layers to compute an accurate sparse $\ell^1$ code. We show that the number of layers can be reduced considerably with homotopy algorithms.

**Homotopy continuation**    Homotopy continuation algorithms introduced in Osborne et al. (2000), minimize the $\ell^1$ Lagrangian (1) by progressively decreasing the Lagrange multiplier. This optimization path is opposite to ISTA and FISTA since it begins with a very sparse initial solution whose sparsity is progressively reduced, similarly to matching pursuit algorithms (Davis et al., 1997; Donoho & Tsaig, 2008). Homotopy algorithms are particularly efficient if the final Lagrange multiplier $\lambda_*$ is large and thus produces a very sparse optimal solution. We shall see that it is the case for classification.

Homotopy proximal gradient descents (Xiao & Zhang, 2013) are implemented with an exponentially decreasing sequence of Lagrange multipliers $\lambda_n$ for $n \leq N$. Jiao, Jin and Lu (Jiao et al., 2017) have introduced an Iterative Soft Thresholding Continuation (ISTC) algorithm with a fixed number of iterations per threshold. To compute a positive sparse code, we replace the soft thresholding by a ReLU proximal projector, with one iteration per threshold, over $n \leq N$ iterations:

$$\alpha_n = \rho(\alpha_{n-1} + D^t(\beta - D\alpha_{n-1}) - \lambda_n) \text{ with } \lambda_n = \lambda_{\max} \left( \frac{\lambda_{\max}}{\lambda_*} \right)^{-n/N} \tag{5}$$

By adapting the proof of (Jiao et al., 2017) to positive codes, the next theorem proves in a more general framework that if $N$ is sufficiently large and $\lambda_{\max} \geq \|D^t\beta\|_\infty$ then $\alpha_n$ converges exponentially to the optimal positive sparse code.

LISTA algorithm (Gregor & LeCun, 2010) and its more recent version ALISTA (Liu et al., 2019) accelerate the convergence of proximal algorithms by introducing an auxiliary matrix $W$, which is adapted to the statistics of the input and to the properties of the dictionary. Such an auxiliary matrix may also improve classification accuracy. We study its influence by replacing $D^t$ by an arbitrary matrix $W^t$ in (5). Each column $W_m$ of $W$ is normalized by $|W_m^t D_m| = 1$. A generalized ISTC is defined for any dictionary $D$ and any auxiliary $W$ by:

$$\alpha_n = \rho(\alpha_{n-1} + W^t(\beta - D\alpha_{n-1}) - \lambda_n) \text{ with } \lambda_n = \lambda_{\max} \left( \frac{\lambda_{\max}}{\lambda_*} \right)^{-n/N} \tag{6}$$

If $W = D$ then we recover the original ISTC algorithm (5) (Jiao et al., 2017). Figure 2 illustrates a neural network implementation of this generalized ISTC algorithm over $N$ layers, with side connections. Let us introduce the mutual coherence of $W$ and $D$

$$\widetilde{\mu} = \max_{m \neq m'} |W_{m'}^t D_m|$$

The following theorem gives a sufficient condition on this mutual coherence and on the thresholds so that $\alpha_n$ converges exponentially to the optimal sparse code. ALISTA (Liu et al., 2019) is a particular case of generalized ISTC where $W$ is optimized in order to minimize the mutual coherence $\widetilde{\mu}$. In Section 4.1 we shall optimize $W$ jointly with $D$ without any analytic mutual coherence minimization like in ALISTA.

**Theorem 3.1** *Let $\alpha^0$ be the $\ell^0$ sparse code of $\beta$ with error $\|\beta - D\alpha^0\| \leq \sigma$. If its support $s$ satisfies*

$$s\,\widetilde{\mu} < 1/2 \tag{7}$$

*then thresholding iterations (6) with*

$$\lambda_n = \lambda_{\max}\, \gamma^{-n} \geq \lambda_* = \frac{\|W^t(\beta - D\alpha^0)\|_\infty}{1 - 2\gamma\widetilde{\mu}s} \tag{8}$$

*define an $\alpha_n$, whose support is included in the support of $\alpha^0$ if $1 < \gamma < (2\widetilde{\mu}s)^{-1}$ and $\lambda_{\max} \geq \|W^t\beta\|_\infty$. The error then decreases exponentially:*

$$\|\alpha_n - \alpha^0\|_\infty \leq 2\,\lambda_{\max}\, \gamma^{-n} \tag{9}$$

The proof is in Appendix A of the supplementary material. It adapts the convergence proof of Jiao et al. (2017) to arbitrary auxiliary matrices $W$ and positive sparse codes. If we set $W$ to minimize the mutual coherence $\widetilde{\mu}$ then this theorem extends the ALISTA exponential convergence result to the noisy case. It proves exponential convergence by specifying thresholds for a non-zero approximation error $\sigma$.

However, one should not get too impressed by this exponential convergence rate because the condition $s\widetilde{\mu} < 1/2$ only applies to very sparse codes in highly incoherent dictionaries. Given a dictionary $D$, it is usually not possible to find $W$ which satisfies this hypothesis. However, this sufficient condition is based on a brutal upper bound calculation in the proof. It is not necessary to get an exponential convergence. Next section studies learned dictionaries for classification on ImageNet and shows that when $W = D$, the ISTC algorithm converges exponentially although $s\mu(D) > 1/2$. When $W$ is learned independently from $D$, with no mutual coherence condition, we shall see that the algorithm may not converge.

## 4 IMAGE CLASSIFICATION

The goal of this work is to construct a deep neural network model which is sufficiently simple to be analyzed mathematically, while reaching the accuracy of more complex deep convolutional networks on large classification problems. This is why we concentrate on ImageNet as opposed to MNIST or CIFAR. Next section shows that a single $\ell^1$ sparse code in a learned dictionary improves considerably the classification performance of a scattering representation, and outperforms AlexNet on ImageNet [1]. We analyze the influence of different architecture components. Section 4.2 compares the convergence of homotopy iterated thresholdings with ISTA and FISTA.

### 4.1 IMAGE CLASSIFICATION ON IMAGENET

ImageNet 2012 (Russakovsky et al., 2015) is a challenging color image dataset of 1.2 million training images and 50,000 validation images, divided into 1000 classes. Prior to convolutional networks, SIFT representations combined with Fisher vector encoding reached a Top 5 classification accuracy of 74.3% with multiple model averaging (Sánchez & Perronnin, 2011). In their PyTorch implementation, the Top 5 accuracy of AlexNet and ResNet-152 is 79.1% and 94.1% respectively[2].

The scattering transform $Sx$ at a scale $2^J = 16$ of an ImageNet color image is a spatial array of $14 \times 14$ of 1539 channels. If we apply to $Sx$ the same MLP classifier as in AlexNet, with 2 hidden layers of size 4096, ReLU and dropout rate of 0.3, the Top 5 accuracy is 65.3%. We shall use the same AlexNet type MLP classifier in all other experiments, or a linear classifier when specified. If we first apply to $Sx$ a 3-layer SLE network of 1x1 convolutions with ReLU and then the same MLP then the accuracy is improved by 14% and it reaches AlexNet performance (Oyallon et al., 2017). However, there is no mathematical understanding of the operations performed by these three layers, and the origin of the improvements, which partly motivates this work.

The sparse scattering architecture is described in Figure 3. A $3 \times 3$ convolutional operator $L$ is applied on a standardized scattering transform to reduce the number of scattering channels from 1539 to 256. It includes $3.5\,10^6$ learned parameters. The ISTC network illustrated in Figure 2 has $N = 12$ layers with ReLU and no batch normalization. A smaller network with $N = 8$ has nearly the same classification accuracy but the ISTC sparse coding does not converge as well, as explained in Section 4.2. Increasing $N$ to 14 or 16 has little impact on accuracy and on the code precision.

---

[1]Code to reproduce experiments is available at `https://github.com/j-zarka/SparseScatNet`
[2]Accuracies from `https://pytorch.org/docs/master/torchvision/models.html`

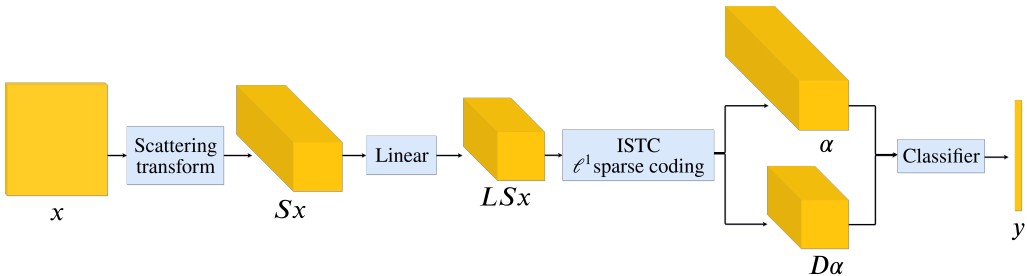

Figure 3: Two variants of the image classification architecture: one where the input for the classifier is the sparse code $\alpha$, and the other where the reconstruction $D\alpha$ is the input for the classifier.

The sparse code is first calculated with a $1 \times 1$ convolutional dictionary $D$ having $2048$ vectors. Dictionary columns $D_m$ have a spatial support of size $1$ and thus do not overlap when translated. It preserves a small dictionary coherence so that the iterative thresholding algorithm converges exponentially. This ISTC network takes as input an array $LSx$ of size $14 \times 14 \times 256$ which has been normalized and outputs a code $\alpha^1$ of size $14 \times 14 \times 2048$ or a reconstruction $D\alpha^1$ of size $14 \times 14 \times 256$. The total number of learned parameters in $D$ is about $5 \, 10^5$. The output $\alpha^1$ or $D\alpha^1$ of the ISTC network is transformed by a batch normalization, and a $5 \times 5$ average pooling and then provided as input to the MLP classifier. The representation is computed with $4 \, 10^6$ parameters in $L$ and $D$, which is above the $2.5 \, 10^6$ parameters of AlexNet. Our goal here is not to reduce the number of parameters but to structure the network into well defined mathematical operators.

Table 1: Top 1 and Top 5 accuracy on ImageNet with a same MLP classifier applied to different representations: Fisher Vectors (Perronnin & Larlus, 2015), AlexNet (Krizhevsky et al., 2012), Scattering with SLE (Oyallon et al., 2019), Scattering alone, Scattering with ISTC for $W = D$ which outputs $\alpha^1$, or which outputs $D\alpha^1$, or which outputs $\alpha^1$ with unconstrained $W$.

|  | Fisher Vectors | AlexNet | Scat. + SLE | Scat. alone | Scat.+ ISTC $\alpha^1$ , $W = D$ | Scat.+ ISTC $D\alpha^1$ , $W = D$ | Scat.+ ISTC $\alpha^1$ , $W \neq D$ |
|---|---|---|---|---|---|---|---|
| Top1 | 55.6 | 56.5 | 57.0 | 42.0 | 59.2 | 56.9 | 62.8 |
| Top5 | 78.4 | 79.1 | 79.6 | 65.3 | 81.0 | 79.3 | 83.7 |

If we set $W = D$ in the ISTC network, the supervised learning jointly optimizes $L$, the dictionary $D$ with the Lagrange multiplier $\lambda_*$ and the MLP classifier parameters. It is done with a stochastic gradient descent during $160$ epochs using an initial learning rate of $0.01$ with a decay of $0.1$ at epochs $60$ and $120$. With a sparse code in input of the MLP, it has a Top 5 accuracy of $81.0\%$, which outperforms AlexNet.

If we also jointly optimize $W$ to minimize the classification loss, then the accuracy improves to $83.7\%$. However, next section shows that in this case, the ISTC network does not compute a sparse $\ell^1$ code and is therefore not mathematically understood. In the following we thus impose that $W = D$.

The dimension reduction operator $L$ has a marginal effect in terms of performance. If we eliminate it or if we replace it by an unsupervised PCA dimension reduction, the performance drops by less than $2\%$, whereas the accuracy drops by almost $16\%$ if we eliminate the sparse coding. The number of learned parameters to compute $\alpha^1$ then drops from $4 \, 10^6$ to $5 \, 10^5$. The considerable improvement brought by the sparse code is further amplified if the MLP classifier is replaced by a much smaller linear classifier. A linear classifier on a scattering vector has a (Top 1, Top 5) accuracy of $(26.1\%, 44.7\%)$. With a ISTC sparse code with $W = D$ in a learned dictionary the accuracy jumps to $(51.6\%, 73.7\%)$ and hence improves by nearly $30\%$.

The optimization learns a relatively large factor $\lambda_*$ which yields a large approximation error $\|LSx - D\alpha^1\| / \|LSx\| \approx 0.5$, and a very sparse code $\alpha^1$ with about $4\%$ non-zero coefficients. The sparse approximation $D\alpha^1$ thus eliminates nearly half of the energy of $LS(x)$ which can be interpreted as non-informative "clutter" removal. The sparse approximation $D\alpha^1$ of $LSx$ has a small

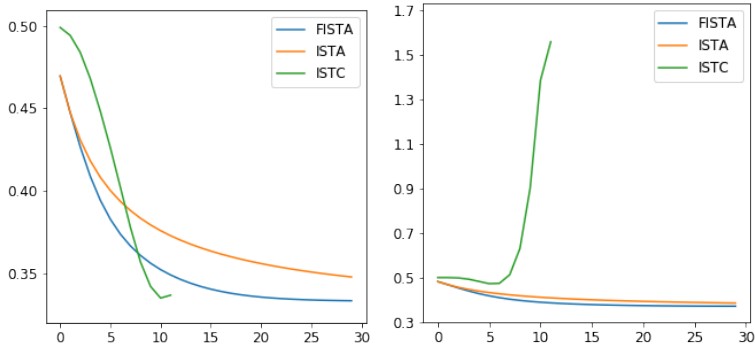

Figure 4: Value of $\mathcal{L}(\alpha_n) = \frac{1}{2}\|D\alpha_n - \beta\|^2 + \lambda_* \|\alpha_n\|_1$ versus the number of iterations $n$, for ISTC with $W = D$, ISTA and FISTA on the left, and for ISTC with $W \neq D$, ISTA and FISTA on the right.

dimension $14 \times 14 \times 256$ similar to AlexNet last convolutional layer output. If the MLP classifier is applied to $D\alpha^1$ as opposed to $\alpha^1$ then the accuracy drops by less than $2\%$ and it remains slightly above AlexNet. Replacing $LSx$ by $D\alpha^1$ thus improves the accuracy by $14\%$. The sparse coding projection eliminates "noise", which seems to mostly correspond to intra-class variabilities while carrying little discriminative information between classes. Since $D\alpha^1$ is a sparse combination of dictionary columns $D_m$, each $D_m$ can be interpreted as "discriminative features" in the space of scattering coefficients. They are optimized to preserve discriminative directions between classes.

### 4.2 CONVERGENCE OF HOMOTOPY ALGORITHMS

To guarantee that the network can be analyzed mathematically, we verify numerically that the homotopy ISTC algorithm computes an accurate approximation of the optimal $\ell^1$ sparse code in (1), with a small number of iterations.

When $W = D$, Theorem 3.1 guarantees an exponential convergence by imposing a strong incoherence condition $s\,\mu(D) < 1/2$. In our classification setting, $s\mu(D) \approx 60$ so the theorem hypothesis is clearly not satisfied. However, this incoherence condition is not necessary. It is derived from a relatively crude upper bound in the proof of Appendix A.1. Figure 4 left shows numerically that the ISTC algorithm for $W = D$ minimizes the Lagrangian $\mathcal{L}(\alpha) = \frac{1}{2}\|D\alpha - \beta\|^2 + \lambda_* \|\alpha\|_1$ over $\alpha \geq 0$, with an exponential convergence which is faster than ISTA and FISTA. This is tested with a dictionary learned by minimizing the classification loss over ImageNet.

If we jointly optimize $W$ and $D$ to minimize the classification loss then the ImageNet classification accuracy improves from $81.0\%$ to $83.7\%$. However, Figure 4 right shows that the generalized ISTC network outputs a sparse code which does not minimize the $\ell^1$ Lagrangian at all. Indeed, the learned matrix $W$ does not have a minimum joint coherence with the dictionary $D$, as in ALISTA (Liu et al., 2019). The joint coherence then becomes very large with $s\tilde{\mu} \approx 300$, which prevents the convergence. Computing $W$ by minimizing the joint coherence would require too many computations.

To further compare the convergence speed of ISTC for $W = D$ versus ISTA and FISTA, we compute the relative mean square error $\mathrm{MSE}(x, y) = \|x - y\|^2/\|x\|^2$ between the optimal sparse code $\alpha^1$ and the sparse code output of 12 iterations of each of these three algorithms. The MSE is 0.23 for FISTA and 0.45 for ISTA but only 0.02 for ISTC. In this case, after 12 iterations, ISTC reduces the error by a factor 10 compared to ISTA and FISTA.

### 5 CONCLUSION

This work shows that learning a single dictionary is sufficient to improve the performance of a predefined scattering representation beyond the accuracy of AlexNet on ImageNet. The resulting deep convolutional network is a scattering transform followed by a positive $\ell^1$ sparse code, which are well defined mathematical operators. Dictionary vectors capture discriminative directions in

the scattering space. The dictionary approximations act as a non-linear projector which removes non-informative intra-class variations.

The dictionary learning is implemented with an ISTC network with ReLUs. We prove exponential convergence in a general framework that includes ALISTA. A sparse scattering network reduces the convolutional network learning to a single dictionary learning problem. It opens the possibility to study the network properties by analyzing the resulting dictionary. It also offers a simpler mathematical framework to analyze optimization issues.

### ACKNOWLEDGMENTS

This work was supported by the ERC InvariantClass 320959, grants from Région Ile-de-France and the PRAIRIE 3IA Institute of the French ANR-19-P3IA-0001 program. We thank the Scientific Computing Core at the Flatiron Institute for the use of their computing resources. We would like to thank Eugene Belilovsky for helpful discussions and comments.

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

## A  APPENDIX

### A.1  PROOF OF THEOREM 3.1

Let $\alpha^0$ be the optimal $\ell^0$ sparse code. We denote by $\mathcal{S}(\alpha)$ the support of any $\alpha$. We also write $\rho_\lambda(a) = \rho(a - \lambda)$. We are going to prove by induction on $n$ that for any $n \geq 0$ we have $\mathcal{S}(\alpha_n) \subset \mathcal{S}(\alpha^0)$ and $\|\alpha_n - \alpha^0\|_\infty \leq 2\lambda_n$ if $\lambda_n \geq \lambda_*$.

For $n = 0$, $\alpha_0 = 0$ so $\mathcal{S}(\alpha_0) = \emptyset$ is indeed included in the support of $\alpha^0$ and $\|\alpha_0 - \alpha^0\|_\infty = \|\alpha^0\|_\infty$. To verify the induction hypothesis for $\lambda_0 = \lambda_{\max} \geq \lambda_*$, we shall prove that $\|\alpha^0\|_\infty \leq 2\lambda_{\max}$.

Let us write the error $w = \beta - D\alpha^0$. For all $m$

$$\alpha^0(m)W_m^t D_m = W_m^t \beta - W_m^t w - \sum_{m \neq m'} \alpha^0(m')\, W_m^t D_{m'}.$$

Since the support of $\alpha^0$ is smaller than $s$, $W_m^t D_m = 1$ and $\widetilde{\mu} = \max_{m \neq m'} |W_m^t D_{m'}|$

$$|\alpha^0(m)| \leq |W_m^t \beta| + |W_m^t w| + s\, \widetilde{\mu}\, \|\alpha^0\|_\infty$$

so taking the max on $m$ gives:

$$\|\alpha^0\|_\infty (1 - \widetilde{\mu} s) \leq \|W^t \beta\|_\infty + \|W^t w\|_\infty$$

But given the inequalities

$$
\begin{aligned}
\|W^t \beta\|_\infty &\leq \lambda_{\max} \\
\|W^t w\|_\infty &\leq \lambda_{\max}(1 - 2\gamma\widetilde{\mu} s) \\
\frac{(1 - \gamma\widetilde{\mu} s)}{(1 - \widetilde{\mu} s)} &\leq 1 \quad \text{since } \gamma \geq 1 \text{ and } (1 - \widetilde{\mu} s) > 0
\end{aligned}
$$

we get

$$\|\alpha^0\|_\infty \leq 2\lambda_{\max} = 2\lambda_0$$

Let us now suppose that the property is valid for $n$ and let us prove it for $n + 1$. We denote by $D_{\mathcal{A}}$ the restriction of $D$ to vectors indexed by $\mathcal{A}$. We begin by showing that $\mathcal{S}(\alpha_{n+1}) \subset \mathcal{S}(\alpha^0)$. For any $m \in \mathcal{S}(\alpha_{n+1})$, since $\beta = D\alpha^0 + w$ and $W_m^t D_m = 1$ we have

$$
\begin{aligned}
\alpha_{n+1}(m) &= \rho_{\lambda_{n+1}}(\alpha_n(m) + W_m^t(\beta - D\alpha_n)) \\
&= \rho_{\lambda_{n+1}}(\alpha^0(m) + W_m^t(D_{\mathcal{S}(\alpha^0) \cup \mathcal{S}(\alpha_n) - \{m\}}(\alpha^0 - \alpha_n)_{\mathcal{S}(\alpha^0) \cup \mathcal{S}(\alpha_n) - \{m\}} + w))
\end{aligned}
$$

For any $m$ not in $\mathcal{S}(\alpha^0)$, let us prove that $\alpha_{n+1}(m) = 0$. The induction hypothesis assumes that $\mathcal{S}(\alpha_n) \subset \mathcal{S}(\alpha^0)$ and $\|\alpha^0 - \alpha_n\|_\infty \leq 2\lambda_n$ with $\lambda_n \geq \lambda_*$ so:

$$
\begin{aligned}
I &= |\alpha^0(m) + W_m^t(D_{\mathcal{S}(\alpha^0) \cup \mathcal{S}(\alpha_n) - \{m\}}(\alpha^0 - \alpha_n)_{\mathcal{S}(\alpha^0) \cup \mathcal{S}(\alpha_n) - \{m\}} + w)| \\
&\leq |W_m^t(D_{\mathcal{S}(\alpha^0)}(\alpha^0 - \alpha_n)_{\mathcal{S}(\alpha^0)})| + |W_m^t w| \quad \text{since } \mathcal{S}(\alpha_n) \subset \mathcal{S}(\alpha^0) \text{ and } \alpha^0(m) = 0 \text{ by assumption.} \\
&\leq \widetilde{\mu} s\|\alpha^0 - \alpha_n\|_\infty + \|W^t w\|_\infty
\end{aligned}
$$

Since we assume that $\lambda_{n+1} \geq \lambda_*$, we have

$$\|W^t w\|_\infty \leq (1 - 2\gamma\widetilde{\mu} s)\lambda_{n+1}$$

and thus

$$I \leq \widetilde{\mu} s\|\alpha^0 - \alpha_n\|_\infty + \|W^t w\|_\infty \leq \widetilde{\mu} s 2\lambda_n + \lambda_{n+1}(1 - 2\gamma\widetilde{\mu} s) \leq \lambda_{n+1}$$

since $\lambda_n = \gamma\lambda_{n+1}$.

Because of the thresholding $\rho_{\lambda_{n+1}}$, it proves that $\alpha_{n+1}(m) = 0$ and hence that $\mathcal{S}(\alpha_{n+1}) \subset \mathcal{S}(\alpha^0)$.

Let us now evaluate $\|\alpha^0 - \alpha_{n+1}\|_\infty$. For any $(\alpha_1, \alpha_2, \lambda)$, a soft thresholding satisfies

$$|\rho_\lambda(\alpha_1 + \alpha_2) - \alpha_1| \leq \lambda + |\alpha_2|$$

so:

$$
\begin{aligned}
|\alpha_{n+1}(m) - \alpha^0(m)| \quad &\leq \quad \lambda_{n+1} + |W_m^t(D_{\mathcal{S}(\alpha^0) \cup \mathcal{S}(\alpha_n) - \{m\}}(\alpha^0 - \alpha_n)_{\mathcal{S}(\alpha^0) \cup \mathcal{S}(\alpha_n) - \{m\}})| + |W_m^t w| \\
&\leq \quad \lambda_{n+1} + \widetilde{\mu} s \|\alpha^0 - \alpha_n\|_\infty + \|W^t w\|_\infty \\
&\leq \quad \lambda_{n+1} + \widetilde{\mu} s 2\lambda_n + \lambda_{n+1}(1 - 2\gamma\widetilde{\mu}s) = 2\lambda_{n+1}
\end{aligned}
$$

Taking a max over $m$ proves the induction hypothesis.

