# OpenReview forum: "Deep Network Classification by Scattering and Homotopy Dictionary Learning"
_ICLR.cc/2020/Conference — Accept (Poster)_

### Official Review · AnonReviewer1 · 2019-10-22
**Official Blind Review #1**

**Rating:** 6

**Review:**

The work considers a new architecture for artificial neural networks to be used in image classification tasks. The network combines several promising previous research directions into an interesting approach (to the best of my knowledge, the architecture is novel).

In the first step, the input representation (i.e., image) is processed using deep scattering spectrum. This is an operator proposed by Mallat (2012) and it is known for extracting robust features with properties such as translation invariance and Lipschitz continuity of the operator mapping. As a result, such a representation is likely to be more stable to perturbations and noise in the inputs.
The scattering operator of the second and higher orders tends to produce a large number of coefficients. Such a representation is, in general, sparse with many of the higher order coefficients equal to zero. To lower the dimension of the scattering representation, the architecture employs a linear projection operator. For example, one can take principal component analysis and project to a sub-space retaining most of the variation present in the data.
The first two blocks (scattering and linear projection) are unsupervised and, thus, kept fixed during learning.

The third block in the architecture aims at finding a sparse coding dictionary that will take into account instances and labels. It builds a dictionary using a convolutional neural network and finds sparse coding using previous work on dictionary learning (e.g., Donoho & Elad, 2006: Marial et al., 2011; Jiao et al., 2017 etc). The algorithm works by computing the sparse encoding vector (see Eq. 1) in the forward pass and updating the convolutional parameters as well as sparsity controlling hyperparameter in the backward step. To speed up the convergence, the authors rely on homotopy iterated thresholding illustrated in Figure 2.

The approach is evaluated on ImageNet and empirical results demonstrate that the removal of the sparse encoding block amounts to a significant performance degradation. The results also establish a minor improvement in the accuracy as a result of adding a linear projection matrix (i.e., principal component analysis applied to scattering coefficients). Overall, the network shows promising performance on ImageNet by doing better than AlexNet. The result is not yet 'competitive' with ResNets but it might be worth to pursue this direction of research in the future.

The work is properly structured with well organized materials from previous work. The clarity, however, could be improved in several places. In particular, the last paragraph in Section 3.1 does not explain the algorithm clearly. The confusing part is how to update the dictionary (i.e., convolutional network) parameters. One might infer from the current materials that first the problem in Eq. (1) is solved to find \alpha and that solution is fixed. Then, for that setting of the sparse encoding vector one will train the network parameters. Section 3.2 then givens an iterative procedure in Eq. (3) and Figure 2, which suggest that in a forward pass the dictionary representation is computed using some setting of parameters and \alpha is updated per Eq. (3). Following this, the gradient of the convolutional parameters is computed (\alpha_{n + 1} is differentiated, which means that the gradient depends on other \alpha iterates). I believe that this is the crux of the methodological/conceptual contribution and requires proper explanation with proper illustration of the forward-backward pass.

In Section 3.2 (homotopy iterated thresholding and ALISTA), there is a matrix W which comes ad-hoc. There should be some motivation and gentle introduction. At the moment, it is completely justified to ask why one needs this matrix and why the approach would not work without Proposition 3.1. Please add some discussion and make sure things are properly motivated and gently introduced. This will also place the theoretical contribution in the proper context and strengthen the work.

Just below Figure 2, I fail to follow how the number of layers N relates to the iterative algorithm? Does it mean that you would actually have blocks per each \alpha_n for some N indices (this again refers to previous comment on clarity)?

Can you please use \ell_1 or l_1 notation for sparse dictionary coding. The current symbol reads as 1 to the power of 1 and it is very confusing (never seen it before in the context of sparse encodings).

**Experience Assessment:**

I have read many papers in this area.

**Review Assessment: Checking Correctness Of Derivations And Theory:**

I did not assess the derivations or theory.

**Review Assessment: Checking Correctness Of Experiments:**

I assessed the sensibility of the experiments.

**Review Assessment: Thoroughness In Paper Reading:**

I read the paper at least twice and used my best judgement in assessing the paper.

---

> ### Author Response · Authors · 2019-11-14
> **Answer to review #1**
>
> First of all, thank you very much for your review. Notations have been adjusted according to the reviewer’s suggestions. Please find below answers to your comments.
>
> Reviewer: The last paragraph in Section 3.1 does not explain the algorithm clearly. The confusing part is how to update the dictionary parameters. [...] I believe that this is the crux of the methodological/conceptual contribution and requires proper explanation with proper illustration of the forward-backward pass. [...] Just below Figure 2, I fail to follow how the number of layers N relates to the iterative algorithm? Does it mean that you would actually have blocks per each \alpha_n for some N indices (this again refers to previous comment on clarity)?
>
> We rewrote Section 3.1 to address this point. Section 3 now clearly separates the sparse coding from the dictionary learning stage. We explain that the sparse coding is implemented as a deep network calculation. The dictionary learning is then performed by minimizing the classification loss with stochastic gradient descent. The number of layers in the network is equal to the number of iterations used to approximate the sparse code: we clarify it in the caption of Figure 2. We explain now that during training, the forward pass approximates the sparse code with respect to the current dictionary, and the backward pass updates the dictionary through a stochastic gradient descent step.
>
>
> Reviewer: In Section 3.2 (homotopy iterated thresholding and ALISTA), there is a matrix W which comes ad-hoc. There should be some motivation and gentle introduction. At the moment, it is completely justified to ask why one needs this matrix (W) and why the approach would not work without Proposition 3.1.
>
> The reviewer is right. We now explain the role of this auxiliary matrix to accelerate the convergence of LISTA and ALISTA. Theorem 3.1 gives a general framework to study the role of such matrices, by giving a sufficient condition for exponential convergence. In Section 4 we consider the case where W=D which corresponds to the original ISTC algorithm, and a general case where W is freely optimized.

---

### Official Review · AnonReviewer3 · 2019-10-23
**Official Blind Review #3**

**Rating:** 8

**Review:**


## Summary

The paper proposes an interpretable architecture for image classification based on a scattering transform and sparse dictionary learning approach. The scattering transform acts as a pre-trained interpretable feature extractor that does not require data. A sparse dictionary on top of this representation (the scattering coefficients) is learnt to minimize the classification error. The authors cast the dictionary learning as a classical CNN learning approach and implement an efficient solution via homotopy learning (given that some assumptions are fulfilled). The scattering transform approach is not new (as the authors mention in the paper, it was published in Oyallon et al., 2019). The main novelty comes from applying a previously published dictionary learning approach (as the authors mention in the paper, it was published in Jiao et al., 2017) on top to boost the performance. As a second contribution, the authors extend the exponential convergence proof of ISTC (Jiao et al., 2017) and ALISTA (Liu et al., 2019). In the experiments, they show that the proposed architecture, despite its simplicity, outperform AlexNet in the ImageNet classification problem.

## Comments

The paper is well written and the exposition is clear. The main motivation of the paper is to propose an interpretable architecture with similar performance to black box deep learning architectures. To do so, the authors put together:

- A scattering transform feature extractor: Unlike I am missing something, this is exactly what was previously proposed in (Oyallon et al., 2019).
- A dictionary learning on top: This seems to be the biggest novelty of the paper. This component allows to boost the performance of the previously proposed architecture. However, this approach has been previously explored in the literature (Mahdizadehaghdam et al. 2018), the authors just apply it on top the extracted features. The justification of the paper lies in that previous dictionary learning approaches did not scale (convergence too slow), and so the authors use a different method recently published in (Jiao et al., 2017).

This allow the authors to apply the method to bigger datasets ImageNet, and keep the performance above AlexNet.

Generalizing the convergence results of ALISTA and ISTC is a nice contribution. However, my main concern is with respect the novelty of the rest of the paper. The authors do not propose a substantially different approach, rather they apply the same approach (an scalable  dictionary learning method already published in Jiao et al., 2017) on top of some extracted features (scattering coefficients)  to a different datasets. The problem with accepting the paper is that changing the dataset/dictionary learning method/features to compare with, you get a different paper, and so, in my opinion, the impact of this publication is limited.

Also, given that the paper main point is the interpretability of the proposed method wrt to black-box deep learning methods, I think the authors should include recent references to the active field of interpretability in the deep neural network community.

**Experience Assessment:**

I do not know much about this area.

**Review Assessment: Checking Correctness Of Derivations And Theory:**

I assessed the sensibility of the derivations and theory.

**Review Assessment: Checking Correctness Of Experiments:**

I assessed the sensibility of the experiments.

**Review Assessment: Thoroughness In Paper Reading:**

I read the paper at least twice and used my best judgement in assessing the paper.

---

> ### Author Response · Authors · 2019-11-14
> **Answer to review #3**
>
> First of all, thank you very much for your review. Please find below answers to your comments.
>
> Reviewer: The scattering transform approach is not new (Oyallon et al. 2019). The main novelty comes from applying a previously published dictionary learning approach on top to boost the performance [...] - A scattering transform feature extractor: Unlike I am missing something, this is exactly what was previously proposed in (Oyallon et al., 2019).
>
> We modified the introduction to clarify this confusion. The scattering transform is a predefined representation (no learning) published in 2013 (Bruna et al.) (cited). Since then, several papers (which are cited) have obtained state-of-the-art results with a scattering among all predefined and unsupervised learning representations. It also performs nearly as well as learned deep networks on relatively simple image datasets (MNIST, CIFAR). Oyallon et al., 2019, did not introduce the scattering transform but have shown that an important classification gap is observed on ImageNet, which can be bridged by adding a CNN on top of scattering. However, Oyallon paper does not provide an explanation.
>
> Understanding this gap is about understanding the nature of the information learned by deep neural networks for image classification. The goal of this paper is to address this problem through a model which is as simple as possible. The paper indeed shows that learning a single sparse coding dictionary matrix is sufficient to reach AlexNet performance and bridge the gap. The learned dictionary matrix captures the discriminative information needed to reach a high performance. This is the main novelty of the paper. It provides a simplified mathematical model of CNN for image classification.
>
>
> Reviewer: [...] and so the authors use a different method recently published in (Jiao et al., 2017). This allow the authors to apply the method to bigger datasets ImageNet, and keep the performance above AlexNet. Generalizing the convergence results of ALISTA and ISTC is a nice contribution. However, my main concern is with respect the novelty of the rest of the paper. The authors do not propose a substantially different approach, rather they apply the same approach (an scalable dictionary learning method already published in Jiao et al., 2017) on top of some extracted features (scattering coefficients) to a different datasets.
>
> We have rewritten Section 3.1 to address this misunderstanding. The ISTC and ALISTA algorithms, as well as ISTA or FISTA are algorithms which find the minimum of a sparse coding loss for a given dictionary D. This is very different from dictionary learning, which is a much more complex non-convex problem that optimizes the dictionary D. Our contribution is a dictionary learning algorithm, which incorporates a generalized ISTC into a deep network in order to optimize the dictionary D through stochastic gradient descent.
>
> Section 3.1 now explains the principles of sparse coding and dictionary learning for classification and clearly differentiates the two problems. Section 3.2 is devoted to sparse coding algorithms only. The goal is to reduce the number of network layers for computational efficiency. We show that by using ISTC algorithms we obtain a more efficient dictionary learning algorithm in a deep neural network.
>
>
> Reviewer: - A dictionary learning on top: This seems to be the biggest novelty […] However, this approach has been previously explored in the literature (Mahdizadehaghdam et al. 2018) [...] The justification of the paper lies in that previous dictionary learning approaches did not scale (convergence too slow). [...] The problem with accepting the paper is that changing the dataset/dictionary learning method/features to compare with, you get a different paper, and so, in my opinion, the impact of this publication is limited.
>
> We do not agree with this conclusion. We now explain more clearly in the introduction that all previously dictionary learning approaches, including Mahdizadehaghdam et al., 2018, consist of cascading *many dictionaries* and involve the learning of many intermediate matrices. There is no indication that these operators actually compute sparse $\ell^1$ codes. As a result they are mathematically not understood and so complex that they could not be applied to ImageNet.
>
> The introduction explains that this is the first paper which shows that one can reach deep network performance by learning a *single dictionary matrix D* from a predefined representation (scattering transform) and we prove that it actually performs a sparse $\ell^1$ coding. This provides a simple learning model that scales to large datasets, which we believe can have a high impact.
>
>
> Reviewer: The authors should include recent references to the active field of interpretability
>
> To be more precise and avoid confusion between mathematical analysis and interpretability, we have replaced “mathematical interpretability” by “mathematical understanding”.

---

### Official Review · AnonReviewer2 · 2019-10-25
**Official Blind Review #2**

**Rating:** 8

**Review:**

The paper proposes a network architecture composed of three interpretable components followed by a simple MLP classifier. It first applies a scattering transform followed by a learned linear projection (to reduce dimensionality). A sparse representation of these coefficients is then obtained using dictionary learning. The projection, dictionary and MLP classifier are jointly trained to minimize the classification loss. Results show that the model outperforms AlexNet on the Imagenet benchmark.

The paper is well written. The main contribution of the work is to present an architecture with composed of mathematically interpretable components achieving very high empirical performance. I find that these results are very significant.

The second contribution is to propose a dictionary learning algorithm that uses ISTC and can be trained with gradient descent.  I think that it would be interesting to add an ablation showing the change in performance by changing N (the number of iterations in the ISTC net). Also, it would make sense to run FISTA or ISTA unrolls to see if the benefits of the faster convergence also affect classification performance.

It would be good to add to Table 1 the number of trainable parameters of each variant.

I find it a bit confusing to refer to the setting in which W is learned as ALISTA, as to me ALISTA implies using analytical W. This is clear later in the text (and makes sense from a computational standpoint). Would be good to clarify it early in the text.

Finally the paper presents a proof of exponential convergence for ALISTA in the noisy case. While this is an interesting result, it is not very closely linked to the main focus of the work.


**Experience Assessment:**

I have published one or two papers in this area.

**Review Assessment: Checking Correctness Of Derivations And Theory:**

I assessed the sensibility of the derivations and theory.

**Review Assessment: Checking Correctness Of Experiments:**

I assessed the sensibility of the experiments.

**Review Assessment: Thoroughness In Paper Reading:**

I read the paper thoroughly.

---

> ### Author Response · Authors · 2019-11-14
> **Answer to review #2**
>
> First of all, thank you very much for your review. Please find below answers to your comments.
>
> Reviewer: I think that it would be interesting to add an ablation showing the change in performance by changing N (the number of iterations in the ISTC net).
>
> This is now included in the text of Section 4.1. Reducing the number of iterations to N=8 preserves the accuracy but the sparse code does not converge to an $\ell^1$ sparse code and thus can not be interpreted. For N = 14 or 16 the convergence accuracy is nearly the same with marginal convergence improvements.
>
>
> Reviewer: Also, it would make sense to run FISTA or ISTA unrolls to see if the benefits of the faster convergence also affect classification performance.
>
> Figure 4, left shows that the sparse code produced by ISTC (W = D) converges to the same sparse code produced by ISTA/FISTA, but ISTA/FISTA require many more iterations. We tried to apply ISTA unrolled with N=12 iterations using the same dictionary learning procedure as for ISTC. It provides a classification accuracy of 79.5% which is above AlexNet but it provides a poor approximation of an $\ell^1$ sparse code, as shown in Figure 4.
>
>
> Reviewer: It would be good to add to Table 1 the number of trainable parameters of each variant.
>
> We have now incorporated the number of trainable parameters to compute the sparse scattering representation, in Section 4.1, depending upon the configuration. This is incorporated in the text with a comparison with AlexNet.
>
>
> Reviewer: I find it a bit confusing to refer to the setting in which W is learned as ALISTA, as to me ALISTA implies using analytical W. This is clear later in the text (and makes sense from a computational standpoint). Would be good to clarify it early in the text.
>
> To clarify all this, we define a general ISTC algorithm for an arbitrary W. For W which minimizes the mutual coherence we get ALISTA. We explain that Section 4.1 considers the case of an arbitrary W which does not correspond to ALISTA.
>
>
> Reviewer: Finally the paper presents a proof of exponential convergence for ALISTA in the noisy case. While this is an interesting result, it is not very closely linked to the main focus of the work.
>
> We now explain more clearly that Theorem 3.1 gives a sufficient condition for exponential convergence of the general ISTC algorithm, which is used in all experiments of Section 4. If W has a minimum mutual convergence with D then we get a convergence result for ALISTA in the noisy case. This result also considerably simplifies ALISTA proof and shows that it is a homotopy algorithm. However, we agree with the reviewer that it is not used in the numerics so we have removed this from the introduction, to concentrate on the main focus of the work.

---

### Author Response · Authors · 2019-11-14
**Some general clarifications**

We are very grateful to all the reviewers for their comments which were important for us to improve the paper. We have modified the introduction to clarify the goal and contributions of the paper, especially for Reviewer #3.

We partly rewrote Section 3.1 to better differentiate sparse coding problems from dictionary learning following remarks of Reviewer #1 and #3. We also tried to better highlight the specificities of different sparse coding algorithms in Section 3.2 to address remarks of all reviewers. To clarify the role of W, we describe the ISTC algorithm for a general W and explain how it relates to the original ISTC and ALISTA for different values of W, and the role of W.

For simplicity, we have restricted the sparse codes to positive codes, so that the network is implemented with ReLU and a bias as in standard CNNs. It also improved classification accuracies by about 0.5%. The Section 4 was adapted to these modifications.

---

### Decision · Program_Chairs · 2019-12-19

**Decision:**

Accept (Poster)

**Comment:**

After the rebuttal period the ratings on this paper increased and it now has a strong assessment across reviewers. The AC recommends acceptance.